# Impact of Spa Therapy on Symptoms and Quality of Life in Post-COVID-19 Patients with Chronic Conditions

**DOI:** 10.3390/jcm13175091

**Published:** 2024-08-27

**Authors:** Maria Costantino, Valentina Giudice, Mario Farroni, Francesco Marongiu, Francesco De Caro, Amelia Filippelli

**Affiliations:** 1Department of Medicine, Surgery, and Dentistry, University of Salerno, 84081 Baronissi, Italy; vgiudice@unisa.it (V.G.); fdecaro@unisa.it (F.D.C.); afilippelli@unisa.it (A.F.); 2University Hospital “San Giovanni di Dio e Ruggi d’Aragona”, 84121 Salerno, Italy; 3Non-Profit Association F.I.R.S.Thermae (Interdisciplinary Training, Researches and Spa Sciences), 80078 Pozzuoli, Italyfrancescomarongiu@firsthermae.org (F.M.)

**Keywords:** post COVID, Spa therapy, QoL, vaccination

## Abstract

**Background:** With limited pharmacological interventions, post-COVID-19 condition is a clinical challenge, and supplementary therapies are essential for symptom relief and enhancing quality of life (QoL). In our prospective observational study, we aimed to evaluate the impact of Salus per aquam (Spa) therapy on post-COVID-19 symptoms and QoL in individuals who suffer from chronic joint, musculoskeletal, skin, and/or respiratory conditions. **Methods:** A total of 159 individuals undergoing Spa therapy were enrolled, and 78 of them had post-COVID-19 symptoms, assessed using Visual Analogue Scale (VAS) and modified British Medical Research Council Questionnaire (mMRC-DS scales), as well as the Short Form 36 Health Status Survey (SF-36) questionnaire for QoL. **Results:** Spa therapy significantly reduced most post-COVID-19 symptoms, especially chronic fatigue, pain, brain fog, and persistent cough (all *p* < 0.05), as well as physical (+72%) and emotional (+66%) limitations. When stratified by sex, males showed a greater improvement from baseline, while females consistently displayed a higher amelioration in all QoL dimensions. Moreover, full vaccination with 3–4 doses significantly protected against SARS-CoV-2 re-infections and post-COVID-19 development (*p* < 0.05). **Conclusions:** Spa therapy demonstrated effectiveness in mitigating post-COVID-19 symptoms and enhancing QoL in patients suffering from chronic diseases.

## 1. Introduction

The novel coronavirus SARS-CoV-2 is responsible for the COVID-19 pandemic and causes a disease with various clinical presentations, from asymptomatic cases to severe acute respiratory distress syndrome and death [1,2]. Moreover, approximately 10% of infected individuals experience a condition known as post-COVID-19 condition [3], defined as the persistence of symptoms after four weeks since acute infection resolution in a subject who has recovered from COVID-19 and tested negative. This syndrome is likely caused by direct virus-induced tissue damage or virus-triggered autologous immune responses against organs and tissues [4,5]. Several risk factors have been proposed, including heart failure, neurological diseases, female sex, and chronic diseases, such as rheumatic diseases, type 2 diabetes, and obesity [6,7,8,9,10]. Post-COVID-19 condition can present with a wide and varied symptomatology, observed weeks or months after the acute phase of COVID-19 infection [11,12,13,14], and can include shortness of breath, chronic fatigue, sleep disorders, visual problems, joint pain, and the so-called brain-fog (difficulty in thinking or concentration) [15]. Symptoms could also differ based on variants, as the ancestral virus infection is more frequently linked to anosmia, dysgeusia, and hearing problems, while its alpha variant is more commonly associated with myalgia, insomnia, brain-fog, anxiety, and depression [15]. In addition, many other symptoms have been reported, such as persistent cough, tachycardia, arrhythmias, nausea, vomiting, tinnitus, ear pain, urticaria, or cutaneous rashes [16,17,18]. Patients with persistent symptoms frequently complain of reduced Quality of Life (QoL) [19,20]. To date, no effective pharmacological therapies are available to reduce the incidence and duration of post-COVID-19, although full vaccination with three doses is the only recognized protective factor [21,22].

Salus per aquam (Spa) therapy, also known as crenotherapy, shows therapeutic effects because of intrinsic properties of natural mineral waters often rich in minerals, such as sulfur, magnesium, calcium, and potassium, that are used in various forms, including baths, mud treatments, inhalations, and showers. Combination of heat, hydrostatic pressure, and chemical composition of waters can induce muscle relaxation, improve blood circulation, and reduce inflammation [23,24,25,26]. In rheumatic and musculoskeletal conditions, Spa therapy can alleviate pain and improve mobility, while it can exert anti-inflammatory and relaxing effects in dermatological conditions, such as psoriasis, eczema, and dermatitis, or can improve tissue damage recovery, circulation, and pain relief after surgery or trauma. In addition, inhalation of minerals-enriched steam is used to treat chronic respiratory conditions like bronchitis and sinusitis [23]. 

Because of these anti-inflammatory, antioxidant, analgesic, and muscle relaxant properties, Spa therapy could represent an alternative approach for post-COVID-19 treatment [23,24,26,27,28,29,30]. Indeed, Spa therapy has beneficial effects in chronic conditions, including chronic fatigue syndrome (CFS), chronic low back pain, or skin diseases, because it can reduce pain and skin symptoms, while improving physical function and quality of life (QoL) [31,32,33]. Crenotherapy is frequently prescribed with clinical benefits for treatment of chronic articular disorders, a risk factor of post-COVID-19 development [6,7,8,34,35,36,37,38,39,40]. Moreover, balneotherapy is proposed as a rehabilitation option to improve fatigue and muscle pain, two symptoms that frequently characterize post-COVID-19 condition [41,42].

Based on these considerations, we aimed to investigate the efficacy of Spa therapies for post-COVID-19 symptom relief in patients suffering from chronic joint, musculoskeletal, skin, and/or respiratory diseases requiring Spa treatments. Additionally, the impact on QoL of this therapy was assessed and subsequently stratified by sex. 

## 2. Materials and Methods

### 2.1. Population and Study Design

A total of 160 Caucasian subjects (mean age ± SD, 55 ± 14.8 years old; M/F, 50 (31%)/110 (69%); mean BMI (kg/m^2^) ± SD, 25 ± 4) were included in this prospective observational study from three Italian spas: “Stufe di Nerone” in Bacoli, Naples (N = 70; data collection from April 2023 to January 2024); “Giardini Poseidon” in Ischia, Naples (N = 60; data collection from April 2023 to October 2024); and “Terme dei Colli Asolani” in Pieve del Grappa, Treviso (N = 30; data collection from April 2023 to January 2024). These subjects had a diagnosis of chronic joint, musculoskeletal, skin, and/or respiratory diseases (arthro-rheumatic or/and musculoskeletal or dermatological diseases, N = 119, 74%; respiratory or/and otolaryngological diseases, N = 36, 23%; arthro-rheumatic or/and musculoskeletal diseases + respiratory or/and otolaryngological diseases or + dermatological diseases, N = 5, 3%) and received a prescription from a general practitioner or specialist for a 2-week therapeutic Spa cycle (balneotherapy and/or inhalation). Balneotherapy alone was performed on 30 patients at “Terme dei Colli Asolani”, 60 at “Giardini Poseidon”, and 30 at “Terme Stufe di Nerone”, while inhalation therapy alone on 36 subjects and balneotherapy + inhalation therapy on 4 at “Terme Stufe di Nerone”. Balneotherapy was administered individually as a single bath, consisting of full-body immersion (head excluded) in a bath containing spa mineral waters at a temperature of 37–38 °C for 20 min. After treatment, subjects were covered with a blanket and rested for 30 min lying or reclined in a comfortable position. A complete cycle of balneotherapy lasted 12 days and consisted of 12 consecutive baths (once daily) in a single tub, in the morning and preferably under fasting conditions. Balneotherapy was administered for treatment of chronic arthro-rheumatic, musculoskeletal, and dermatological diseases. Inhalation crenotherapy cycle consisted of twelve 10 min applications of spa mineral waters as direct jet plus 12 aerosols, administered after a 10 min rest in between. Inhalation crenotherapy was employed for treatment of chronic inflammation and/or irritation of upper and lower respiratory tracts and otolaryngological diseases. A timeline with recruitment, treatment, and data collection was included in Appendix A. This study was approved by our local Ethics Committee “Campania Sud”, Naples, Italy (approval No. 44 r.p.s.o./2023), in accordance with the Declaration of Helsinki. A specific case report form (CRF) was used to collect patients’ data. All enrolled subjects met the following inclusion criteria: age ≥ 18 years; history of arthro-rheumatic, musculoskeletal, respiratory, otolaryngological, and dermatological diseases; written informed consent. Exclusion criteria were: presence of acute clinical conditions, cancer, autoimmune diseases, and absence of signed written informed consent.

### 2.2. Anti-SARS-CoV-2 Vaccination Status

Anti-SARS-CoV-2 vaccination status was investigated on the entire cohort, and 141 were vaccinated, while 19 not, with 74% of vaccinated individuals (N = 104) and 95% of non-vaccinated patients (N = 18) reporting at least one documented infection by molecular test (*p* = 0.042). Vaccinated subjects had a significantly lower incidence of repeated SARS-CoV-2 infections compared to non-vaccinated patients (N = 15 vs. N = 8; 14% vs. 44%, respectively; *p* = 0.003). A total of 396 doses were administered, with 80% of mRNA- and 20% of viral vector-based vaccines. Those who received 3–4 doses had a significantly lower infection rate (74%) and post-COVID-19 incidence (60%) compared to those who received 1–2 doses (81%) (*p* = 0.04).

### 2.3. Post-COVID-19 Condition

Post-COVID-19 condition diagnosis was made according to current guidelines [3,11,12,13,14], defined as continuation or new onset of symptoms 3 months after the initial SARS-CoV-2 infection, lasting for at least 2 months with no other explanation, and made by qualified health professionals before Spa therapy initiation in 78 subjects (64.5%). Mean time between COVID-19 infection and Spa therapy start in 121 subjects (76.1%) was 11.8 ± 9.1 months (mean ± standard deviation [SD]).

### 2.4. Symptom and QoL Assessment

Severity of post-COVID-19 symptoms was assessed using the Visual Analogue Scale (VAS), ranging from 0 to 4 (0 = absent, 1 = mild, 2 = moderate, 3 = severe, 4 = extremely severe) [43]. Respiratory symptoms were measured applying the modified British Medical Research Council Questionnaire (mMRC-DS), ranging from 0 to 4 (0 = “I only get breathless with strenuous exercise”; 1 = “I get short of breath when hurrying on the level or walking up a slight hill”; 2 = “I walk slower than people of the same age on the level because of breathlessness or have to stop for breath when walking at my own pace on the level”; 3 = “I stop for breath after walking about 100 yards or after a few minutes on the level”; 4 = “I am too breathless to leave home” or “I am breathless when dressing”) [44]. Impact of Spa treatments on QoL was assessed using the “Short Form 36 Health Status Survey” questionnaire (SF-36), a generic measurement of health values not related to age, treatment, or diseases [45,46,47,48,49], and is the most widely used, reliable, and internationally recognized tool for QoL measurement. This questionnaire was culturally adapted and validated in Italy as part of the IQOLA project [49]. The SF-36 is comprised of eight health dimensions (or different health aspects) with multiple-choice questions for a total of 36 items. The eight health domains are: Physical Functioning (PF); Role Limitations due to Physical health (RLP); Role Limitations due to Emotional problems (RLE); Energy and Fatigue (EF); Emotional Well-Being (EWB); Social Functioning (SF); Body Pain (P); and General Health perceptions (GH). Each item is scored on a scale ranging from 0 (poor health) to 100 (optimal health) [46,48,49]. To evaluate satisfaction on Spa treatments, an unbalanced ordinal numeric scale was also employed (1 = definitely satisfied; 2 = very satisfied; 3 = quite satisfied; 4 = neither satisfied nor dissatisfied; 5 = quite unsatisfied; and 6 = definitely unsatisfied), as suggested elsewhere [50].

For investigation of safety and tolerability of Spa therapy, undesired events during treatment were recorded in CRF, and exacerbation of symptoms reported before treatment and new symptom onset during Spa cycle were registered. 

### 2.5. Statistical Analysis

Demographics characteristics of study population were assessed by descriptive analysis. Continuous variables were presented as mean ± SD, and two-group comparisons were performed using an unpaired or paired *t*-test for normally distributed data, while Wilcoxon’s signed-rank test was used for non-normal distributed variables. Categorical variables were analyzed using χ^2^ test. Percent variation (Δ%) of VAS scores of symptoms or other values before and after Spa therapy was calculated as follows: Δ% = [(VAS-score after − VAS-score before)/VAS-score before] × 100. 

Sample size was calculated by assuming a delta T0-T1 of the mean score of the SF-36 questionnaire equal to 10, an α error at 0.05, and a β error at 0.20, and at least 125 should have been enrolled in this observational study. Cohen’s d was used to quantify effect size for SF-36 questionnaire scores between patients with post-COVID-19 condition and subjects who were never infected, before and after Spa therapy. Cohen’s d was calculated by subtracting mean scores of the reference group (never-infected subjects) from those of the comparison group (patients with post-COVID-19 condition), and by dividing these differences by pooled standard deviations (SD pooled) of the two groups [51]. A positive Cohen’s d value indicated that the comparison group had a higher mean compared to the reference group, while a negative value suggested a lower mean in the comparison group. Standard interpretation criteria were: d = 0.2 was considered as a small effect; d = 0.5 as a medium effect; and d ≥ 0.8 as a large effect. Multiple logistic regression was also performed to investigate associations between prescribed Spa therapy, sex, or anti-COVID-19 vaccination status and clinical features and QoL dimensions. Data collection and analysis were carried out using the SPSS 23.0 statistics package. A *p* value < 0.05 was considered statistically significant.

## 3. Results

### 3.1. Characteristics at Enrollment

From total enrolled patients (N = 160), 159 of them completed a Spa therapeutic cycle, while one person who had post-COVID-19 exited from the study for personal reasons. Therefore, a total of 159 Caucasian subjects were included in further analysis and were stratified based on the presence of post-COVID-19 condition. From the total cohort, 121 subjects (76.1%, females, N = 88 and males, N = 33) contracted SARS-CoV-2 infection, with 78 (64.5%) exhibiting at least one post-COVID-19 symptoms, as summarized in Table 1 with related VAS scores. Patients with post-COVID-19 condition were predominantly females (N = 56, 72%), had a mean age of 55 ± 13.8 years old, and a mean BMI of 26 ± 4 kg/m^2^. Three subjects experienced undesired events during Spa therapy, including dizziness on the first day of treatment, cough, and fatigue at the end treatment; however, despite these symptoms, those subjects completed the Spa cycle.

### 3.2. Effects of Spa Therapy on Post-COVID-19 and Disease-Specific Symptoms

The most prevalent post-COVID-19 symptom was chronic fatigue (N = 52, 67%) with a mean VAS score of 3.2 ± 0.8, followed by muscle pain (N = 37, 47%) with a mean VAS score of 3.1 ± 0.9, and joint pain (N = 32, 41%) with a mean VAS score of 2.0 ± 0.8. Other common symptoms included brain fog, persistent cough, and headache. Sex-stratified analysis showed a higher incidence among females, especially for chronic fatigue and muscle pain. After Spa therapy, there was a significant reduction in chronic fatigue, muscle and joint pain, brain fog, persistent cough, headache, chest pain, dyspnea, sore throat, persistent loss of smell, nausea, and itching (Table 1). For both sexes, subjects experienced substantial symptom improvement after Spa treatment (Table 2). Moreover, Spa therapy significantly improved disease-specific symptoms, especially in females, as reported in those subjects who did not develop post-COVID-19 condition (N = 43; F/M, 74%/26%; mean age ± SD, 50.1 ± 16.4 years old; range, 24–74 years; median age, 56.5 years old; mean BMI ± SD, 23.5 ± 3.5) (Table 3). In addition, symptom improvements in subjects with post-COVID-19 condition were also compared between those patients who received balneotherapy alone and those who performed inhalation therapy alone, to investigate the clinical efficacy of each intervention (Appendix A). Overall, these two types of Spa therapies induced similar clinical benefits, without significant differences. 

### 3.3. QoL Improvement in Post-COVID-19 Patients

QoL was assessed using the SF36 questionnaire, and post-COVID-19 patients had a marked reduction in QoL dimensions compared to non-infected individuals (N = 38) (Table 4). After Spa therapy, significant improvements versus baseline were observed in all QoL dimensions, especially physical (Δ%, +72%) and emotional (Δ%, +66%) limitations (Table 4 and Figure 1A,B). Similarly, subjects who have never had COVID-19 showed a significant improvement in QoL dimensions after Spa cycle (RLP and RLE, Δ%, +42% and Δ%, +31%, respectively; all *p* < 0.05). When stratified by sex, males reported a significant amelioration in all QoL dimensions, except EWB (Δ%, +18%) (*p* > 0.050) (Table 5 and Figure 1C). Similarly, a significant improvement in all QoL dimensions was reported in females (all *p* < 0.05), especially for RLE (Δ%, +73%) and RLP (Δ%, +71%) dimensions, because females started with lower RLE and RLP values at Spa initiation (55 ± 46 and 50 ± 37 vs. 48 ± 44 and 45 ± 41, males vs. females, respectively; all *p* < 0.001), as well as in other QoL dimensions (Table 5 and Figure 1D). Moreover, despite scores after treatment being similar between the two groups, females tended to have a greater improvement compared to males.

Moreover, Cohen’s d effects were also studied between patients with post-COVID-19 and never-infected subjects, showing values ranging between −0.18 and −0.72 before Spa and from −0.18 to −0.55 after Spa therapy, with variable effect sizes (Appendix A). Before Spa therapy, the post-COVID-19 group had significantly lower scores, reflecting a poorer QoL compared to the control group with never-infected individuals, suggesting that post-COVID-19 condition had a substantial negative impact on these aspects of life. After Spa therapy, a positive impact on all QoL dimensions was described, and never-infected individuals tended to have higher scores compared to post-COVID-19 patients across all SF-36 items, suggesting that QoL was overall higher in these subjects, with the most significant differences in emotional functioning and pain.

Finally, a logistic regression analysis was performed to assess the association between prescribed Spa therapy, sex, or anti-COVID-19 vaccination status and clinical features and QoL dimensions in long-COVID-19 patients (Appendix A). We showed that females more frequently were referred to balneotherapy, and RLE after Spa therapy was positively associated (estimate, 0.0569; 95% confidential interval [CI], 0.008539–0.1205; *p* = 0.0420). Conversely, EF before Spa and SF after Spa were associated with female sex. No associations were described with studied variables and anti-COVID-19 vaccination status.

### 3.4. Satisfaction of Spa Treatment 

Regarding Spa treatment satisfaction, 47.5% and 36% of patients were “definitely satisfied” or “very satisfied”, respectively, using an unbalanced ordinal numeric scale, and 14% of subjects were “quite satisfied”. The remaining 2.6% of cases were “neither satisfied nor dissatisfied” or “quite dissatisfied”.

## 4. Discussion

Post-COVID-19 condition is a clinical challenge, and supplementary therapies are essential for symptom relief and enhancing QoL. In our prospective observational study, we showed a potential benefit of Spa therapy in reducing most post-COVID-19 condition symptoms, especially chronic fatigue, pain, brain fog, and persistent cough, as well as physical and emotional limitations, thus improving QoL. We also stratified our results by sex, and males showed a greater improvement from baseline, while females consistently displayed a higher amelioration in all QoL dimensions. As an additional finding, we confirmed that full vaccination with 3–4 doses significantly protects against SARS-CoV-2 re-infections and post-COVID-19 condition development. Therefore, Spa therapy could be a complementary effective approach for post-COVID-19 condition COVID treatment, especially in patients with chronic diseases, together with full vaccination, emphasizing its importance in managing post-COVID complications.

Crenotherapy, or Salus per aquam (Spa) therapy, is an alternative therapeutic option for various disorders, including musculoskeletal, arthro-rheumatic, respiratory, dermatological, and otorhinolaryngology conditions, because of its antioxidant, anti-inflammatory, chondroprotective, muscle relaxant, and decontracting properties [24,25,26,27,28,29,52]. Clinical benefits depend on the physical properties of treatments used and on the unique chemical characteristics of natural mineral waters, such as those employed in our study, rich in combined sulfur, bicarbonate ions, sodium chloride, bromine, and iodine. Indeed, sodium chloride exhibits antiseptic properties, iodine enhances metabolic and cellular activity, while bromide and calcium provide soothing and pain-relieving effects on mucous membranes. Inhalation treatment with bicarbonate–sulphate–alkaline–-earth carbonated waters is beneficial for upper respiratory tract chronic conditions in humans and mouse models, as tobacco smoke-exposed rats show improved alveolar surfactant actions after inhalation crenotherapy with sulphate mineral waters [53,54]. Similarly, Spa treatment with sodium chloride sulphate waters augments ciliary motility in elderly subjects with chronic rhinosinusitis [55], while treatment with inhaled salsojodic mineral waters has vasodilating activity on bronchial mucosa and increases secretory IgA production and muco-ciliary clearance [56,57]. Current hypotheses on mechanisms of action of inhalation Spa treatments propose that glutathione functions are enhanced under Spa therapy, leading to a reduction in oxidative stress, inflammation, and reactive oxygen species generation [57,58]. Balneotherapy with rich-in-sulfur natural mineral water reduces joint pain and cartilage deterioration with increased mobility in rat models of osteoarthritis and in patients, as demonstrated by pain relief and improving physical function and QoL after Spa therapy [59,60]. Similarly, Spa balneotherapy improves antioxidant status and reduces inflammation in patients suffering from lower back pain, knee, and hand osteoarthritis [61]. These clinical benefits might be linked to increased cortisol secretion, and reduced pro-inflammatory (e.g., interleukin-6) and augmented anti-inflammatory (e.g., interleukin-10) cytokine production [62,63]. Due to these extensive therapeutic effects, we explored the clinical impact of balneotherapy and/or inhalation Spa therapies as an alternative treatment strategy for post-COVID-19 condition. In the Italian population, 87.1% of subjects who got COVID-19 syndrome report at least one symptom after 60 days from acute infection, with 55% of them experiencing three or more symptoms, including dyspnea, fatigue, and reduced QoL [18,64,65]. Clinical management of post-COVID-19 remains a challenge, due to the wide range of associated symptoms, often requiring a multidisciplinary approach. In this real-life prospective observational study, we explored the clinical impact of Spa therapy (balneotherapy and/or inhalation cycle) on post-COVID-19 symptoms and QoL. Indeed, previous studies have already proposed a role of complementary therapies for improving post-COVID-19-related symptoms, such as systemic whole-body hyperthermia [66]. On the other hand, Spa therapy has shown beneficial effects in chronic conditions, including CFS, chronic low back pain, skin diseases, or chronic articular disorders [31,32,33]. Therefore, these clinical benefits can be translated to treatment of post-COVID-19, which is frequently characterized by altered physical functions, fatigue, and muscle and articular pain [6,7,8,34,35,36,37,38,39,40,42].

Post-COVID-19 is associated with a wide range of symptoms, as identified by the American Center for Disease Control [11]. In our cohort, general symptoms, including tiredness or fatigue, were the most reported, followed by respiratory, neurological, and musculoskeletal symptoms, regardless of gender. After Spa therapy, a significant improvement documented by reductions in VAS scores was observed in almost all symptoms, including the most complained, such as chronic fatigue, muscle and joint pain, brain fog, cough, and headache. Females greatly benefited from Spa therapies with a significant amelioration of VAS scores for almost all symptoms, except for dysphagia and vomiting. Males with post-COVID-19 who completed the cycle did not experience an improvement in taste disorders, pseudo freezing hands and feet, tinnitus, persistent loss of smell, dysphony, gastroesophageal reflux, nausea, burning sensation of the skin, dysphagia, and dyspnea. These changes may result from a combination of biological and psychological factors, and better clinical benefits of Spa therapy observed in females compared to males could be linked to its antioxidant properties, as documented in previous studies [24,26,28,30,52]. Sex-related modulations of basal cellular redox states and responses to oxidative stress are greater in females [52,67], likely due to protective effects of estrogens and higher expression of genes involved in stress response [67,68]. For example, glutathione peroxidase activities are increased in the kidneys and brains of female mice compared to males, as well as superoxide dismutase activity being more pronounced in the brains and lungs of females [69].

In addition, we have recently shown that crenotherapy with sulfureous mineral water can improve cellular redox state after a 2-weeks’ treatment, with a more marked improvements of clinical symptoms in females with osteoarthritis or degenerative joint disease and Vulgar psoriasis, and a greater reduction in itching in females with high oxidant species at baseline [52]. These differences are likely related also to a higher attention to symptoms and body changes in women compared to men [70,71]. Recent literature indicates that about 50% of individuals who have recovered from SARS-CoV-2 complain about symptoms after weeks or months from primary infection, as also documented in our cohort with a post-COVID-19 incidence of 64% [72]. 

After Spa therapy, symptom relief was accompanied by significant ameliorations in QoL, particularly in limitations related to physical and emotional roles. Despite females with post-COVID-19 condition having a poorer perception of QoL at enrollment compared to males, females showed a greater amelioration after Spa therapy compared to males.

These variations in QoL perception after treatment could be attributed to different coping strategies between genders. Indeed, females often bear the responsibility of managing crises, caring for children, family, and home, rendering them more susceptible to the related psychological burdens and mental health sequelae [71]. Moreover, Spa therapy significantly improved disease-specific symptoms in both sexes, as documented in the subgroup of patients without post-COVID-19. 

Finally, the protective role of vaccination against long COVID development and recurrence of COVID-19 infections was investigated, showing that individuals who have received 3–4 doses had a greater protection compared to those who have received 1–2 doses, with a lower incidence of post-COVID-19 and a reduction in SARS-CoV-2 infection and reinfection rates. Our results strongly support the efficacy of vaccination in post-COVID-19 prevention and COVID-19 recurrence. Therefore, our data added evidence to vaccine safety, supporting its use in general populations and reducing vaccination hesitancy [73,74,75].

Our study has some limitations: (i) limited duration of observation; (ii) a small number of unvaccinated subjects; (iii) the non-randomized nature of this study; and (iv) absence of waiting list.

## 5. Conclusions

In conclusion, Spa therapy could represent an effective alternative therapeutic approach for post-COVID-19 symptom relief and for QoL improvement, especially when this treatment is integrated into a multidisciplinary strategy that also includes vaccination. Thus, Spa therapy can offer valuable support to other medical approaches in managing post-COVID-19 condition. Future perspectives are: (i) investigating long-term effects of spa therapy on long COVID symptoms; (ii) conducting randomized controlled trials to establish causality and determine the most effective spa therapy protocols; and (iii) integrating spa therapy with other therapeutic and complementary approaches to optimize patient outcomes. However, further research in larger prospective cohorts is required to validate our findings.

## Figures and Tables

**Figure 1 jcm-13-05091-f001:**
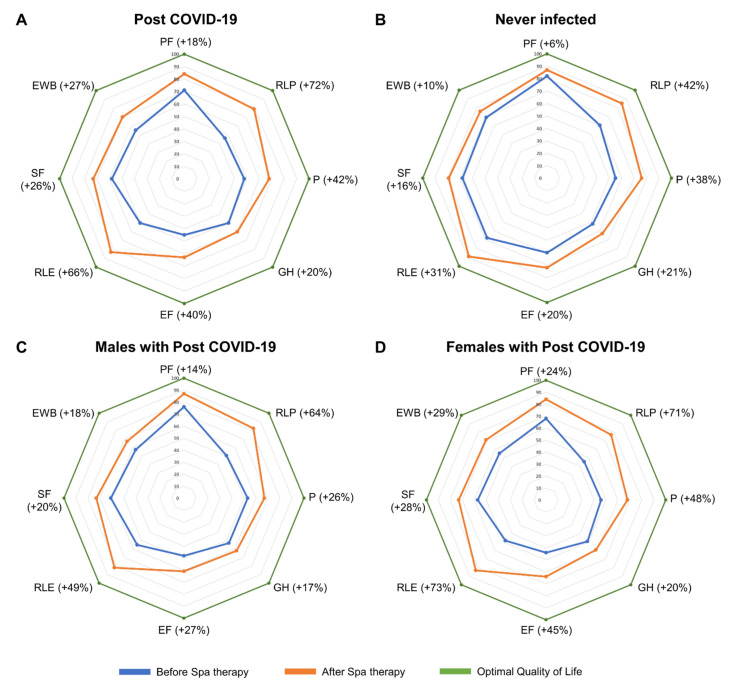
Quality of Life (QoL) dimension improvements after Spa therapy. QoL dimensions determined by the SF-36 questionnaire before (blue line) and after (orange line) Spa treatment in subjects who developed post-COVID-19 (**A**) and who had never experienced SARS-CoV-2 infection (**B**). Optimal QoL is reported in the green line. Patients with post-COVID-19 were also stratified by sex, and QoL dimensions before (blue line) and after (orange line) Spa treatment in males (**C**) and females (**D**) are reported.

**Table 1 jcm-13-05091-t001:** Post-COVID-19 symptoms before and after Spa therapy.

Symptoms (N, %)	Total N = 78 N (%)	Females N = 56 N (%)	Males N = 22 N (%)	*p* Value	VAS Score		*p* Value	Δ%
					Before	After		
Chronic fatigue	52 (67)	39 (70)	13 (59)	0.374	3.2 ± 0.8	1.7 ± 1.1	0.001	−47%
Muscle pain	37 (47)	29 (52)	8 (36)	0.220	3.1 ± 0.9	1.6 ± 1.1	0.001	−48%
Joint pain	32 (41)	23 (41)	9 (41)	0.990	2.0 ± 0.8	1.6 ± 1.0	0.001	−20%
Brain fog	28 (36)	22 (39)	6 (27)	0.320	2.9 ± 1.0	1.7 ± 1.0	0.001	−41%
Persistent cough	24 (31)	16 (29)	8 (36)	0.502	2.8 ± 1.2	0.9 ± 1.1	0.001	−68%
Headache	23 (29)	17 (30)	6 (27)	0.788	2.8 ± 1.0	1.1 ± 1.0	0.001	−61%
Chest pain	14 (18)	8 (14)	6 (27)	0.179	2.3 ± 0.9	0.7 ± 0.9	0.001	−70%
Dyspnea	13 (17)	9 (16)	4 (18)	0.822	2.3 ± 1.3	0.8 ± 0.8	0.003	−65%
Taste disorders	13 (17)	9 (16)	4 (18)	0.822	2.8 ± 0.9	1.2 ± 1.3	0.001	−57%
Pseudo freezing hands and feet	13 (17)	8 (14)	5 (23)	0.368	2.4 ± 1.3	1.0 ± 1.1	0.001	−58%
Tinnitus	13 (17)	8 (14)	5 (23)	0.368	2.9 ± 1.0	1.8 ± 1.0	0.01	−38%
Gastroesophageal reflux	11 (14)	8 (14)	3 (14)	0.941	2.8 ± 0.9	1.0 ± 0.8	0.001	−64%
Sore throat	10 (13)	7 (13)	3 (14)	0.893	2.8 ± 1.1	0.3 ± 0.7	0.001	−89%
Persistent loss of smell	10 (13)	7 (13)	3 (14)	0.893	3.3 ± 1.1	1.0 ± 1.3	0.001	−70%
Burning sensation of the skin	9 (12)	7 (13)	2 (9)	0.672	2.9 ± 0.6	1.0 ± 1.1	0.001	−66%
Hair loss	9 (12)	9 (16)	0 (0)	0.046	2.4 ± 1.2	1.1 ± 0.6	0.01	−54%
Dysphony	8 (10)	6 (11)	2 (9)	0.832	1.8 ± 1.0	0.1 ± 0.4	0.01	−94%
Nausea	7 (9)	5(9)	2 (9)	0.982	3.0 ± 0.6	0.4 ± 0.5	0.001	−87%
Itching	7 (9)	4 (7)	3 (14)	0.367	2.9 ± 1.3	0.9 ± 0.9	0.001	−69%
Earache	6 (8)	6 (11)	0 (0)	0.110	2.8 ± 0.4	1.5 ± 1.0	0.03	−46%
Dysphagia	4 (5)	2 (3.6)	2 (9)	0.320	2.3 ± 1.0	0.3 ± 0.5	0.04	−87%
Vomiting	3 (4)	2 (3.6)	1 (4.5)	0.840	3.3 ± 0.6	0.3 ± 0.6	0.04	−91%
Nasal congestion	2 (3)	1 (1.8)	1 (4.5)	0.488	2.5 ± 0.7	0.5 ± 0.7	0.3	−80%
Irritability	2 (3)	1 (1.8)	1 (4.5)	0.488	2.5 ± 0.7	1.5 ± 0.7	0.707	−40%

**Table 2 jcm-13-05091-t002:** Post-COVID-19 symptoms stratified by sex before and after Spa therapy.

Symptoms	Females with Post-COVID-19 N = 56				Males with Post-COVID-19 N = 22			
	Before	After	Δ%	*p* Value	Before	After	Δ%	*p* Value
Chronic fatigue	3.2 ± 0.8	1.7 ± 1.1	−47%	0.001	3.0 ± 0.8	1.5 ± 1.3	−53%	0.001
Muscle pain	3.1 ± 0.9	1.7 ± 1.1	−45%	0.001	3.1 ± 0.8	1.4 ± 1.1	−55%	0.006
Joint pain	2.9 ± 0.9	1.6 ± 1.1	−45%	0.001	3.2 ± 0.7	1.7 ± 0.9	−47%	0.002
Brain fog	2.9 ± 1.0	1.7 ± 0.9	−41%	0.001	2.7 ± 1.4	1.5 ± 1.4	−44%	0.03
Persistent cough	2.8 ± 1.2	0.8 ± 1.1	−71%	0.001	3.0 ± 1.2	1.1 ± 1.2	−66%	0.01
Headache	2.9 ± 1.0	1.4 ± 1.0	−52%	0.001	2.5 ± 1.0	0.5 ± 0.8	−80%	0.001
Taste disorders	2.6 ± 1.0	1.1 ± 1.3	−58%	0.003	3.3 ± 0.5	1.3 ± 1.5	−67%	0.12
Pseudo freezing hands and feet	2.1 ± 1.2	0.8 ± 0.9	−62%	0.004	2.8 ± 1.3	1.4 ± 1.5	−50%	0.11
Tinnitus	3.0 ± 1.1	1.8 ± 1.0	−40%	0.03	2.8 ± 0.8	2.0 ± 1.0	−29%	0.1
Sore throat	2.6 ± 1.1	0.4 ± 0.8	−85%	0.003	3.3 ± 1.2	0.0 ± 0.0	−100%	0.04
Chest pain	2.5 ± 0.9	1.0 ± 0.9	−60%	0.01	2.0 ± 0.9	0.3 ± 0.8	−85%	0.02
Persistent loss of smell	3.1 ± 1.2	1.6 ± 1.1	−48%	0.01	3.7 ± 0.6	1.0 ± 1.7	−73%	0.09
Dysphony	1.7 ± 1.0	0.2 ± 0.4	−88%	0.03	2.0 ± 1.4	0.0 ± 0.0	−100%	0.3
Gastroesophageal reflux	2.8 ± 1.0	1.0 ± 0.8	−64%	0.001	3.0 ± 0.0	1.0 ± 1.0	−67%	0.07
Nausea	2.8 ± 0.4	0.6 ± 0.5	−79%	0.001	3.5 ± 0.7	0.0 ± 0.0	−100%	0.09
Itching	2.3 ± 1.5	0.8 ± 1.0	−65%	0.01	3.7 ± 0.6	1.0 ± 1.0	−73%	0.02
Hair loss	2.4 ± 1.2	1.1 ± 0.6	−54%	0.01	-	-	-	-
Burning sensation of the skin	3.0 ± 0.6	1.0 ± 1.2	−67%	0.01	2.5 ± 0.7	1.0 ± 1.4	−60%	0.2
Nasal congestion	3	0	−100%	-	2	1	−50%	-
Dysphagia	2.0 ± 1.4	0.0 ± 0.0	−100%	0.3	2.5 ± 0.7	0.5 ± 0.7	−80%	0.3
Earache	2.8 ± 0.4	1.5 ± 1.0	−46%	0.03	-	-	-	-
Irritability	2	1	−50%	-	3	2	−33%	-
Vomiting	3.0 ± 0.0	0.5 ± 0.7	−83%	0.1	4	0	−100%	-
Dyspnea	2.4 ± 1.3	0.8 ± 0.7	−67%	0.02	2.0 ± 1.4	0.8 ± 1.0	−60%	0.08

**Table 3 jcm-13-05091-t003:** Disease-specific symptoms in subjects without post-COVID-19 before and after Spa therapy.

Symptoms (N, %)	Total N = 43 N (%)	VAS Score		*p* Value	Females N = 32			Males N = 11		
		Before	After		Before	After	*p* Value	Before	After	*p* Value
Spontaneous pain	23 (53)	2.8 ± 1.0	1.5 ± 1.4	0.01	2.7 ± 0.9	1.5 ± 1.3	0.01	3.0 ± 0.8	1.3 ± 1.9	0.07
Functional pain	28 (65)	3.3 ± 0.5	2.4 ± 0.9	0.01	3.4 ± 0.5	2.5 ± 0.7	0.01	3.2 ± 0.4	1.8 ± 1.3	0.03
Morning stiffness	22 (51)	3.1 ± 0.9	2.0 ± 1.4	0.01	3.1 ± 0.9	1.9 ± 1.3	0.01	3.5 ± 0.7	3.0 ± 1.4	0.5
Paresthesia	18 (462)	2.7 ± 0.8	1.1 ± 1.2	0.01	2.4 ± 0.8	1.1 ± 1.2	0.01	2.7 ± 0.6	1.0 ± 1.7	0.2
Nasal obstruction	6 (14)	3.3 ± 0.5	2.5 ± 0.5	0.01	3.5 ± 0.6	2.5 ± 0.6	0.01	3.0 ± 0.0	2.5 ± 0.7	0.5
Sneezing	6 (14)	2.2 ± 1.0	1.2 ± 1.0	0.01	2.3 ± 1.2	1.3 ± 1.2	0.01	2.0 ± 1.0	1.0 ± 1.0	0.23
Nasal itching	3 (7)	2.7 ± 0.6	1.7 ± 1.2	0.2	3	3	-	2.5 ± 0.7	1.0 ± 0.0	0.2
Headache	4 (9)	2.3 ± 1.0	1.1 ± 1.2	0.02	2.7 ± 0.6	1.3 ± 1.2	0.06	1	0	-
Cough	3 (7)	2.7 ± 0.6	1.7 ± 1.5	0.4	3.0 ± 0.0	1.5 ± 2.1	0.5	2	2	-

**Table 4 jcm-13-05091-t004:** Quality of Life assessment before and after Spa therapy.

SF-36 Items	Subjects with Post-COVID-19 N = 78				Never-Infected Subjects N = 38				*p* Value *	*p* Value ^#^
	Before	After	Δ%	*p* Value	Before	After	Δ%	*p* Value		
PF	71 ± 23	84 ± 16	+18%	<0.0001	82 ± 18	87 ± 17	+6%	0.0022	0.2052	0.4831
RLP	46 ± 40	79 ± 33	+72%	<0.0001	60 ± 42	85 ± 27	+42%	0.0003	0.3950	0.2664
P	48± 26	68 ± 22	+42%	<0.0001	55± 21	76 ± 17	+38%	<0.0001	0.8427	0.0547
GH	50 ± 20	60 ± 18	+20%	<0.0001	52 ± 15	63 ± 13	+21%	<0.0001	0.9889	0.3541
EF	45 ± 22	63 ± 15	+40%	<0.0001	60 ± 18	72 ± 15	+20%	0.0002	0.0215	0.0514
RLE	50 ± 44	83 ± 27	+66%	<0.0001	68 ± 41	89 ± 26	+31%	0.0057	0.0659	0.7487
SF	58 ± 27	73 ± 22	+26%	<0.0001	68 ± 23	79 ± 19	+16%	0.0121	0.4064	0.5917
EWB	55 ± 21	70 ± 15	+27%	<0.0001	69 ± 18	76 ± 19	+10%	0.0190	0.0134	0.2359

Abbreviations. PF, Physical Functioning; RLP, Role Limitations due to Physical health; RLE, Role Limitations due to Emotional problems; EF, Energy and Fatigue; EWB, Emotional Well-Being; SF, Social Functioning; P, Pain; GH, General Health perceptions. * Mann–Whitney test for before Spa therapy effects between subjects with post-COVID-19 and never infected; ^#^ Mann–Whitney test for after Spa therapy effects between subjects with post-COVID-19 and never infected.

**Table 5 jcm-13-05091-t005:** Quality of Life assessment stratified by sex before and after Spa therapy.

SF-36 Items	Males with Post-COVID-19 N = 22				Females with Post-COVID-19 N = 56				*p* Value *	*p* Value ^#^
	Before	After	Δ%	*p* Value	Before	After	Δ%	*p* Value		
PF	76 ± 21	87 ± 14	+14%	0.0074	64 ± 24	84 ± 17	+24%	<0.0001	0.1540	0.3663
RLP	50 ± 37	82 ± 36	+64%	0.0026	45 ± 41	77 ± 33	+71%	<0.0001	0.5094	0.3594
P	53± 29	67 ± 26	+26%	0.0426	46 ± 25	68 ± 21	+48%	<0.0001	0.3582	0.9222
GH	53 ± 17	62 ± 22	+17%	0.0153	49 ± 21	59 ± 17	+20%	<0.0001	0.5589	0.6435
EF	48 ± 21	61 ± 20	+27%	0.0072	44 ± 22	64 ± 13	+45%	<0.0001	0.5225	0.9845
RLE	55 ± 46	82 ± 29	+49%	0.0083	48 ± 44	83 ± 27	+73%	<0.0001	0.5714	0.7400
SF	61 ± 25	73 ± 25	+20%	0.0229	57 ± 28	73 ± 20	+28%	<0.0001	0.5804	0.6325
EWB	57 ± 19	67 ± 17	+18%	0.0639	55 ± 22	71 ± 15	+29%	<0.0001	0.5938	0.5120

Abbreviations. PF, Physical Functioning; RLP, Role Limitations due to Physical health; RLE, Role Limitations due to Emotional problems; EF, Energy and Fatigue; EWB, Emotional Well-Being; SF, Social Functioning; P, Pain; GH, General Health perceptions. * Mann–Whitney test for before Spa therapy effects between males and females; ^#^ Mann–Whitney test for after Spa therapy effects between males and females.

## Data Availability

Data are contained within this article.

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
