# Peer review of "Impact of Spa Therapy on Symptoms and Quality of Life in Post-COVID-19 Patients with Chronic Conditions"

_jcm, 2024, doi:10.3390/jcm13175091_

Round 1

Reviewer 1 Report

Comments and Suggestions for Authors

Thank you for the opportunity to review the manuscript “Impact of Spa Therapy on Symptoms and Quality of Life in Long COVID Patients with Chronic Conditions” (jcm-3103759).

Authors submitted a prospective observational study, to evaluate the impact of Spa therapy on Long COVID symptoms and QoL in individuals who suffer from chronic joint, musculoskeletal, skin, and/or respiratory condition.

This research contribution is very important. However, it should be revised before publication.

 Introduction:

There is a need to provide further studies that address spa therapy in the context of CFS, pain, etc.

This makes it easier for the authors to discuss this literature in detail in the discussions section. Also greater details about previous studies results are needed than currently provided that builds the case for having conducted the current study. This could also strengthen the discussion, as it is quite common to refer to findings from those studies relative to the current study findings in the discussion and conclusions sections.

Methods:

They are understandable and suitable for the design of the study.

 Discussion/Conclusion:

Please improve Discussion according my mentioned points (Introduction).

It was also mentioned in the conclusion that spa therapy should be part of a multidisciplinary strategy. This is a very important consideration. This point should also be included at the end of the discussion and supported by current literature https://doi.org/10.3390/diseases10040097.

 Also it is important to give implications for further research. What are the concrete recommendations for action?

Comments on the Quality of English Language

Moderate editing of English language required.

Author Response

Comments and Suggestions for Authors

Thank you for the opportunity to review the manuscript “Impact of Spa Therapy on Symptoms and Quality of Life in Long COVID Patients with Chronic Conditions” (jcm-3103759).

Authors submitted a prospective observational study, to evaluate the impact of Spa therapy on Long COVID symptoms and QoL in individuals who suffer from chronic joint, musculoskeletal, skin, and/or respiratory condition.

This research contribution is very important. However, it should be revised before publication.

Response to General Comments. We really appreciate this Reviewer’s feedback on our work, and we hope we have implemented this revised version by following comments and suggestions.

Comment 1. Introduction:

There is a need to provide further studies that address spa therapy in the context of CFS, pain, etc. This makes it easier for the authors to discuss this literature in detail in the discussions section. Also greater details about previous studies results are needed than currently provided that builds the case for having conducted the current study. This could also strengthen the discussion, as it is quite common to refer to findings from those studies relative to the current study findings in the discussion and conclusions sections.

Response to Comment 1. We thank the Reviewer for this helpful comment that has markedly improved our manuscript. We have provided further studies on Spa therapy in chronic fatigue syndrome (CFS), pain, and other related conditions. These studies have been integrated into the introduction for strengthening our study. We have also included additional details about previous published results, that will facilitate a more comprehensive discussion and comparison with our findings in the discussion and conclusions sections.

On page 2, lines 54-73, the following was text was added “Salus per aquam (Spa) therapy, also known as crenotherapy, shows therapeutic effects because of intrinsic properties of natural mineral waters often rich in minerals, such as sulfur, magnesium, calcium, and potassium, that are used in various forms, including baths, mud treatments, inhalations, and showers. Combination of heat, hydro-static pressure, and chemical composition of waters can induce muscle relaxation, im-prove blood circulation, and reduce inflammation [22-23]. In rheumatic and musculo-skeletal conditions, Spa therapy can alleviate pain and improve mobility, while can exert anti-inflammatory and relaxing effects in dermatological conditions, such as psoriasis, eczema, and dermatitis, or can improve tissue damage recovery, circulation, and pain relief after surgery or trauma. In addition, inhalation of minerals-enriched steam is used to treat chronic respiratory conditions like bronchitis and sinusitis [22].

Because of these anti-inflammatory, antioxidant, analgesic, and muscle relaxant properties, Spa therapy could represent an alternative approach for Long COVID treatment [22-28]. Indeed, Spa therapy has beneficial effects in chronic conditions, including chronic fatigue syndrome (CFS), chronic low back pain, or skin diseases, because it can reduce pain and skin symptoms, while improving physical function and quality of life (QoL) [29-31]. Crenotherapy is frequently prescribed with clinical benefits for treatment of chronic articular disorders, a risk factor of Long COVID development [5-7,32-38]. Moreover, balneotherapy is proposed as a rehabilitation option to improve fatigue and muscle pain, two symptoms that frequently characterize Long COVID condition [39,40].”

Comment 2. Methods:

They are understandable and suitable for the design of the study.

Response to Comment 2. We thank the Reviewer for this positive comment on our study design and statistical analysis.

Comment 3. Discussion/Conclusion:

Please improve Discussion according my mentioned points (Introduction).

It was also mentioned in the conclusion that spa therapy should be part of a multidisciplinary strategy. This is a very important consideration. This point should also be included at the end of the discussion and supported by current literature https://doi.org/10.3390/diseases10040097. Also it is important to give implications for further research. What are the concrete recommendations for action?

Response to Comment 3. We thank the Reviewer for this helpful comment, and we have incorporated additional studies related to spa therapy in treatment of CFS, pain, and other relevant conditions as described in Response to Comment 1. These studies provide a more comprehensive background and support to our findings. We have emphasized the importance of including spa therapy as part of a multidisciplinary strategy for treating Long COVID, as you suggested. We have supported this point with current literature, including the source you provided (https://doi.org/10.3390/diseases10040097). This addition highlights the necessity of a holistic approach in managing Long COVID symptoms and aligns with current best practices.

On page 8, lines 267-274, the following text was added “Indeed, previous studies have already proposed a role of complementary therapies for improving Long COVID-related symptoms, such as systemic whole-body hyperthermia [65]. On the other hand, Spa therapy has shown beneficial effects in chronic conditions, including CFS, chronic low back pain, skin diseases, or chronic articular disorders [29-31]. Therefore, these clinical benefits can be translated to treatment of Long-COVID, that is frequently characterized by altered physical functions, fatigue, and muscle and articular pain [5-7,32-38-40].”

We thank the Reviewer for suggesting implications for further research, as we have provided recommendations for future studies.

On page 9, lines 319-323, the following text was added “Future perspectives are: (i) investigating long-term effects of spa therapy on Long COVID symptoms; (ii) conducting randomized controlled trials to establish causality and determine the most effective spa therapy protocols; and (iii) integrating spa therapy with other therapeutic and complementary approaches to optimize patient outcomes.”

Comments on the Quality of English Language

Moderate editing of English language required.

Response. We have carefully reviewed and revised the text to improve clarity, coherence, and overall readability. These revisions have been implemented throughout the manuscript to ensure English language meets required standards.

We appreciate your valuable feedback and believe that these revisions significantly enhanced the quality and relevance of our manuscript.

Thank you for your time and consideration.

Reviewer 2 Report

Comments and Suggestions for Authors

Comments to authors

1) Please define abbreviations at place where first used (example VAS and 17
mMRC-DS scales)

2) Minor writing improvement is needed (typo, space)

3) Better to give some more background about spa therapy in terms of how this therapy work and cases this therapy is used or recommended.

4) Whether outcome is associated with gender of individual, discuss briefly.

5) Whether any tests were done using body fluids (like blood)??? after study was completed.

Comments on the Quality of English Language

Over all english is OK

Author Response

Comments to authors

Comment 1. Please define abbreviations at place where first used (example VAS and 17 mMRC-DS scales).

Response to Comment 1. We thank the Reviewer for this point, and we have carefully checked the abbreviations throughout the manuscript.

Comment 2. Minor writing improvement is needed (typo, space).

Response to Comment 2. We apologize for typos and grammar errors, and we have corrected them in the revised version of this manuscript.

Comment 3. Better to give some more background about spa therapy in terms of how this therapy work and cases this therapy is used or recommended.

Response to Comment 3. We thank the Reviewer for this helpful comment that has markedly improved our manuscript. We have provided more background on Spa therapy in other chronic conditions. These studies have been integrated into the introduction for strengthening our study. We have also included additional details about previous published results, that will facilitate a more comprehensive discussion and comparison with our findings in the discussion and conclusions sections.

On page 2, lines 54-73, the following was text was added “Salus per aquam (Spa) therapy, also known as crenotherapy, shows therapeutic effects because of intrinsic properties of natural mineral waters often rich in minerals, such as sulfur, magnesium, calcium, and potassium, that are used in various forms, including baths, mud treatments, inhalations, and showers. Combination of heat, hydro-static pressure, and chemical composition of waters can induce muscle relaxation, im-prove blood circulation, and reduce inflammation [22-23]. In rheumatic and musculo-skeletal conditions, Spa therapy can alleviate pain and improve mobility, while can exert anti-inflammatory and relaxing effects in dermatological conditions, such as psoriasis, eczema, and dermatitis, or can improve tissue damage recovery, circulation, and pain relief after surgery or trauma. In addition, inhalation of minerals-enriched steam is used to treat chronic respiratory conditions like bronchitis and sinusitis [22].

Because of these anti-inflammatory, antioxidant, analgesic, and muscle relaxant properties, Spa therapy could represent an alternative approach for Long COVID treatment [22-28]. Indeed, Spa therapy has beneficial effects in chronic conditions, including chronic fatigue syndrome (CFS), chronic low back pain, or skin diseases, because it can reduce pain and skin symptoms, while improving physical function and quality of life (QoL) [29-31]. Crenotherapy is frequently prescribed with clinical benefits for treatment of chronic articular disorders, a risk factor of Long COVID development [5-7,32-38]. Moreover, balneotherapy is proposed as a rehabilitation option to improve fatigue and muscle pain, two symptoms that frequently characterize Long COVID condition [39,40].”

Comment 4. Whether outcome is associated with gender of individual, discuss briefly.

Response to Comment 4. We thank the Reviewer for this point, and we have added discussion on impact of Spa therapy on clinical outcomes stratified by gender.

On page 9, lines 299-303, the following text was added “In addition, we have recently shown that crenotherapy with sulfureous mineral water can improve cellular redox state after a 2-weeks treatment, with a more marked improvements of clinical symptoms in females with osteoarthritis or degenerative joint disease and Vulgar psoriasis, and a greater reduction in itching in females with high oxidant species at baseline [50].”

Comment 5. Whether any tests were done using body fluids (like blood)??? after study was completed.

Response to Comment 5. We thank the Reviewer for this question; however, no tests using body fluids, such as blood, were conducted after study completion, as this observational study only aimed at evaluating clinical outcomes and symptoms improvement.

Reviewer 3 Report

Comments and Suggestions for Authors

First of all, I would like to congratulate the authors on conducting and writing an excellent study. The introduction is sufficiently detailed, with a precise description of the symptoms of long COVID syndrome. The structural organization of the paper is commendable, from the study design and patient selection to the choice of questionnaires and statistical analysis. The results are meticulously described and beautifully presented graphically. The discussion and conclusion are in line with the obtained results and similar research. Kudos to the authors for their research.

Author Response

First of all, I would like to congratulate the authors on conducting and writing an excellent study. The introduction is sufficiently detailed, with a precise description of the symptoms of long COVID syndrome. The structural organization of the paper is commendable, from the study design and patient selection to the choice of questionnaires and statistical analysis. The results are meticulously described and beautifully presented graphically. The discussion and conclusion are in line with the obtained results and similar research. Kudos to the authors for their research.

Response to General Comments. Thank you very much for your kind and encouraging words regarding our manuscript. We are delighted to hear that you found our study well-conducted and well-written. We are grateful for your positive feedback and are encouraged by your comments. Your acknowledgment of our efforts motivates us to continue our research with the same level of dedication and rigor.

Round 2

Reviewer 1 Report

Comments and Suggestions for Authors

The authors improved the paper and now it is suitable for publication.

Comments on the Quality of English Language

Minor editing of English language required.

Author Response

We thank the Reviewer for the helpfull comments that have markedly increased the quality of our work.